# Application of Ultrafiltration to Produce Sheep’s and Goat’s Whey-Based Synbiotic Kefir Products

**DOI:** 10.3390/membranes13050473

**Published:** 2023-04-28

**Authors:** Arona Pires, Gözdenur Tan, David Gomes, Susana Pereira-Dias, Olga Díaz, Angel Cobos, Carlos Pereira

**Affiliations:** 1Polytechnic Institute of Coimbra, School of Agriculture, Bencanta, 3045-093 Coimbra, Portugal; arona@esac.pt (A.P.); david@esac.pt (D.G.); sudias@esac.pt (S.P.-D.); 2Departamento de Química Analítica, Nutrición y Bromatología, Facultad de Ciencias, Campus Terra, Universidade de Santiago de Compostela, 27002 Lugo, Spain; olga.diaz.rubio@usc.es (O.D.); angel.cobos@usc.es (A.C.); 3Engineering Faculty, University Süleyman Demirel, Merkez/Isparta 32260, Turkey; 4Centro de Estudos dos Recursos Naturais, Ambiente e Sociedade—CERNAS, 3045-601 Coimbra, Portugal

**Keywords:** cheese whey, ultrafiltration, ovine, caprine, kefir, prebiotics, probiotics

## Abstract

Membrane filtration technologies are the best available tools to manage dairy byproducts such as cheese whey, allowing for the selective concentration of its specific components, namely proteins. Their acceptable costs and ease of operation make them suitable for application by small/medium-scale dairy plants. The aim of this work is the development of new synbiotic kefir products based on sheep and goat liquid whey concentrates (LWC) obtained by ultrafiltration. Four formulations for each LWC based on a commercial kefir starter or traditional kefir, without or with the addition of a probiotic culture, were produced. The physicochemical, microbiological, and sensory properties of the samples were determined. Membrane process parameters indicated that ultrafiltration can be applied for obtaining LWCs in small/medium scale dairy plants with high protein concentration (16.4% for sheep and 7.8% for goats). Sheep kefirs showed a solid-like texture while goat kefirs were liquid. All samples presented counts of lactic acid bacteria higher than log 7 CFU/mL, indicating the good adaptation of microorganisms to the matrixes. Further work must be undertaken in order to improve the acceptability of the products. It could be concluded that small/medium-scale dairy plants can use ultrafiltration equipment to valorize sheep’s and goat’s cheese whey-producing synbiotic kefirs.

## 1. Introduction

The food industry produces high volumes of byproducts, with considerable environmental impacts. However, in most cases, the nutritional content of such byproducts makes them an important subject for valorization [1]. Cheese whey (CW), buttermilk (BM), and second cheese whey (SCW) are the byproducts resulting from the production of cheese, butter, and whey cheeses, respectively. Both CW and BM have excellent nutritional and techno-functional properties [2,3,4,5]. Due to its high organic load, and with an estimated worldwide production of 190 million tons/year, CW represents an opportunity for bioenergy and biochemical production [6]. This byproduct represents 80–90% of the milk volume and contains more than 50% of milk nutrients. It has the following composition (% *w*/*v*) in the case of bovine CW: 4.5–5% lactose, 0.6–0.8% protein, 0.4–0.5% lipids, and 0.4–1% mineral salts [7]. Lactose is responsible for the high Biochemical Oxygen Demand (BOD) of whey (ca. 40–50 g/L). The values of total solids (lactose, proteins, and fat) are even higher in ovine and caprine whey when compared to bovine whey [8]. The potential environmental impact of CW represents a problem for the small/medium-scale dairy companies that do not possess the capacity to produce whey powder (WP), whey protein concentrates (WPC), or other whey derivatives, such as whey protein isolates (WPI) and hydrolysates (WPH). This problem is particularly relevant for producers of traditional cheeses based on sheep and/or goat milk. In the case of these producers, sheep and/or goat CW is often used to produce whey cheeses (WC). However, this option has some disadvantages, particularly the high energy input required to produce WC, its short shelf-life, and the fact that the byproduct of such process, deproteinized whey, also known as SCW, presents the same environmental constraints of CW, which result from its high lactose content [9].

Pires et al. [8] made an extensive review of the potential alternatives to valorize CW or SCW. Very promising developments have been made to convert these byproducts into value-added commodities such as biofuels, bioplastics, bacterial cellulose, food colors and flavors, bioactive peptides, and single-cell proteins, as reported by several authors [2,10,11,12]. The dairy industry manages whey mainly by membrane separation or fermentation [13]. Due to the low protein content of cheese whey, membrane processes allow industries to concentrate and fractionate its components, operating at ambient or mild temperatures that respect their functional properties at a cost-effective production. The main membrane process applied in whey protein products manufacture is ultrafiltration [14]. The use of membrane separation technologies, including UF, in the dairy industry and whey valorization has been extensively reviewed [15,16,17,18,19]. The most valuable commercial products obtained in the dairy industry are those based on the highest protein concentration, usually in the form of dehydrated powders which guarantee a long shelf-life. Most commercial dry products are based on bovine whey proteins; they include concentrates (WPC), with protein contents between 35% and 80% manufactured by ultrafiltration, and isolates (WPI), with higher protein concentrations (more than 90%) that can be obtained by ultrafiltration/diafiltration or ion chromatography [20].

The recent growing interest in caprine products, attributed to their specific nutritional and nutraceutical characteristics, (i.e., lower allergenicity of their proteins and higher content of oligosaccharides), makes the recovery of goat CW an opportunity to transform this byproduct into value-added novel dairy products [21]. The results of cost–benefit and sensitivity analyses show that an integrated process of ultrafiltration/nanofiltration (UF/NF) is economically viable in small/medium cheese producers [20]. Sheep dairy products, other than cheese, are also being developed and are looked at as an opportunity to valorize peripheral rural areas through the transformation of dairy products into competitive value-added commodities [22,23,24].

Concomitantly, increasing awareness of health has increased demand for healthy food which is confirmed by raising market sales of functional foods. Functional foods are defined as foods that, in addition to their nutritional value, contain ingredients that act specifically on body functions associated with the control or reduction of the risk of developing some diseases. One such food is kefir. This product is a natural acidic-alcoholic fermented milk drink traditionally produced by milk fermentation using kefir grains. Kefir grains are composed of a complex population of bacteria and yeasts embedded in a polysaccharide-protein matrix. Lactic acid bacteria, acetic acid bacteria, and yeasts are the dominant microorganisms. Kefir can provide probiotic benefits, such as intestinal microecological balance regulation, and antibacterial and anti-inflammatory activity [25]. A systematic review by Vieira et al. [26] reports that the bioactive compounds more commonly found in kefir were exopolysaccharides, including kefiran, bioactive peptides, and organic acids, especially lactic acid. It has been indicated that the kefir peptides have therapeutic potential for the treatment of Alzheimer’s disease [27]. The existence of a relationship between the improvement in skin parameters and the changes in the intestinal microbial balance after kefir consumption has also been reported [28]. The bioconversion of whey by kefir lactic acid bacteria (LAB) may be effective in reducing obesity and obesity-related diseases [29]. Other authors [30,31,32] report that milk kefir enhances bone microarchitecture and metabolism, has osteoprotective effects, and can be used as a nutritional supplement to accelerate fracture healing. The antimicrobial activity of kefir microorganisms derives from the microorganisms’ capacity to adhere to the intestinal epithelium, preventing the adhesion of pathogens, among other properties. Bacteria and yeasts isolated from kefir have been shown to have in vivo and in vitro antimicrobial activity against enteropathogenic bacteria and spoilage fungi [33].

Some authors developed synbiotic drinks from milk fermented by kefir grains and supplemented by inulin, quinoa flour, or with the addition of specific probiotic bacteria [25,34,35]. However, most available milk-based kefir products on the market are produced without kefir grains; they use commercial kefir cultures [36]. This study will also aim to develop new synbiotic kefir products based on sheep and goat UF concentrated CW, fermented by kefir grains or commercial kefir starter, with and without the addition of a commercial probiotic culture, and supplemented with inulin.

## 2. Materials and Methods

### 2.1. Production of Liquid Whey Concentrates

The sheep and goat CW were supplied by external dairy companies and processed at the dairy pilot plant of Escola Superior Agrária de Coimbra (Coimbra, Portugal). Each type of whey (500 L) was subjected to ultrafiltration (UF) in a pilot plant supplied by Proquiga Biotech SA (A Coruña, Spain), equipped with an organic UF membrane (3838 PVDF/polysulfone) with an effective area of filtration of 7 m^2^ and 10 kDa cut-off, supplied by FipoBiotech, Pontevedra, Spain. The process was carried out at 40–45 °C, at a transmembrane pressure of 3.0–3.5 bar, aiming at a volumetric concentration factor (VCF = Vol. Feed/Vol. Retentate) of 20. The UF concentration step allowed for the obtention of 25 L of goat and 25 L of sheep liquid whey concentrate (LWC). The goat and sheep LWCs obtained were pasteurized (65 °C, 30 min) and then homogenized at 15 MPa using a Rannie™ model Bluetop homogenizer (Copenhagen, Denmark). LWCs were frozen at −25 ± 2 °C until they were used to produce goat and sheep kefir formulations.

### 2.2. Manufacture of Goat and Sheep Kefir Products

Sheep or goat LWCs (approximately 12 L) were thawed under refrigeration at 0 °C for 24 h. Subsequently, the samples were heated to 80 ± 2 °C and 2.5% (*w*/*v*) inulin (Fibruline™, Cosucra, supplied by Induxtra de Suministros, Moita, Portugal) was added, being the mixture homogenized at 15 MPa. The mixtures were then cooled to 38 °C, divided into four portions (3 L each), and inoculated with one of the following cultures:Commercial kefir culture: Exact™ Kefir 1 (CHR Hansen, Hoersholm, Denmark) mesophilic and thermophilic culture (*Debaryomyces hansenii*, *Lactococcus lactis* subsp. *cremoris*, *L. lactis* subsp. *lactis* biovar *diacetylactis*, *L. lactis* subsp. *lactis*, *Leuconostoc* and *Streptococcus thermophilus*) at a concentration of 0.01% (*w*/*v*) (EK);A mixture of a commercial kefir culture (Exact™ Kefir 1, CHR Hansen, Hoersholm, Denmark) and a probiotic culture containing *Bifidobacterium bifidus*-BB12, *Lactobacillus acidophilus*-LA5 and *S. thermophilus* (ABT-5™, CHR Hansen, Denmark) at 0.01% each one (EKABT5);A traditional kefir culture at a concentration of 2.5% *v*/*v* (TK);A mixture of the traditional kefir culture at 2.5% *v*/*v* and probiotic culture (ABT-5™) at 0.01% (TKABT5).

The inoculated goat or sheep LWCs were placed in an incubation chamber (Jenogand™ Y 1000, Copenhagen, Denmark) at 37 °C and the pH and titratable acidity were monitored until the products reached a target pH of 4.5. The fermentation process was stopped by rapid cooling to 20 °C in less than 30 min. Afterward, the fermented products were placed in the refrigeration chamber at 4 ± 2 °C for 12 h, approximately. After this process, the goat and sheep kefir products were stored under refrigeration in a chamber at 0 ± 2 °C for 30 days. Products were evaluated at the 1st, 10th, 20th, and 30th days of storage.

### 2.3. Physicochemical Analysis

The total solids content of kefir products was determined by drying the samples in a Schutzart DIN 40050-IP20 Memmert™ oven (Schwabach, Germany), according to NP 703: 1982 for yogurt [37]. The ash content was determined by incineration of dry samples in an Nabertherm™, model LE 4/11/R6 electric muffle furnace (Bremen, Germany). The fat content was determined by the Gerber method (SuperVario-N Funke Gerber™ centrifuge, Berlin, Germany) according to NP 2105:1983 [38] for sheep kefir, and NP 1923:1987 for goat kefir [39]. The total N content was determined by the Kjeldahl method in the Digestion System 6 1007 Digester Tecator™ (Foss Analytical, Häganäs, Sweden) following the AOAC (1997) standard [40]. To calculate the percentage of protein the conversion factor of 6.38 was used. All analyzes were performed in triplicate.

#### 2.3.1. pH and Titratable Acidity

The pH of products was directly determined with a HI 9025 HANNA Instruments pH meter (Leighton Buzzard, UK), to monitor the evolution of the pH over fermentation, immediately after production and on the 10th, 20th, and 30th days of storage. The titratable acidity, expressed in % lactic acid, was determined by titration using a 0.1 N NaOH solution according to NP 701: 1982 for yogurts [41].

#### 2.3.2. Color Parameters

The color of kefir samples was determined with a Minolta™ Chroma Meter, model CR-200B colorimeter (Tokyo, Japan) calibrated with a white standard (CR-A47: Y = 94.7; x 0.313; y 0.3204). The following conditions were used: illuminant C, 1 cm diameter aperture, 10° standard observer. The color coordinates were measured in the CIEL*a*b* system.

Color difference (ΔEab*) was calculated as:ΔEab* = [(L* − L*^0^)^2^ + (a* − a*^0^)^2^ + (b* − b*^0^)^2^]^1/2^(1)
where L*^0^, a*^0^, and b*^0^ and L*, a*, and b* were the values measured for the samples under comparison. A matrix of ΔEab* values between products was constructed. Three measurements were taken for each sample.

#### 2.3.3. Rheological Properties

The rheological properties of the kefir products were evaluated in a rheometer (Rheostress 1, ThermoHaake™, Thermo Fisher Scientific, Waltham, MA, USA) in oscillatory mode. The measurement system consisted of a cone and plate geometry, C60/Ti-0.052 mm (35 mm diameter and 1° angle). Stress sweep tests were performed at 1 Hz to investigate the rheological linear viscoelastic behavior of the samples. The elastic modulus (G′), the viscous modulus (G″), and the complex viscosity (η*) of the products were evaluated in the range of 0.3 to 6.5 rad/s at 3 Pa.

#### 2.3.4. Texture Parameters

A Stable Micro Systems™ texture analyzer, model TA.XT Express Enhanced was used to perform the texture analysis of the sheep kefir samples and the results were calculated using the Specific Expression PC software, version 1,1,12 (Godalming, Surrey, UK). A TPA-type test was run with a penetration distance of 15 mm at 1 mm/s using an acrylic cylindrical probe with a diameter of 25.4 mm and a height of 38.1 mm.

### 2.4. Microbiological Analysis

The microbial counts of lactic acid bacteria (LAB) of the genera *Lactobacillus* spp. and *Lactococcus* spp. were performed after production and over storage. Lactococci and lactobacilli were enumerated on plates at 37 °C for 48 h on M17 agar (in aerobiosis) and on MRS agar (in anaerobiosis) (Biokar Diagnostics, Beauvais, France), respectively, according to ISO 7889, IDF 117 (2003) [42]. Yeasts were enumerated in plates at 25 °C according to ISO 6611 IDF 94 (2204) [43]. Analyses were carried out in triplicate along with two controls for each medium and results are expressed as log CFU/g of product.

### 2.5. Sensory Analysis

Consumer preference tests were conducted with an untrained panel within 7 days of storage. A hedonic test was performed for the kefir samples in order to evaluate the aroma, texture, flavor, and global appreciation on a scale from 1 to 9 (1 = I don’t like it at all to 9 = I like it very much) using a non-trained panel with 30 members [44]. The members of the panel were also asked to rank the samples according to their preference, from 1-most preferred to 4-less preferred, according to ISO 8587 (1988) [45].

### 2.6. Statistical Analysis

Prior to statistical analysis, normal distribution was evaluated using the Kolmogorov–Smirnov test. The differences among formulations produced with the same LWC were analyzed by one-way ANOVA and the means were compared by Tukey’s post hoc test. The same statistical treatment was performed to study the effect of storage time on the hardness of sheep kefir samples. The differences between each formulation produced with different LWC (sheep or goat) were compared by *t*-test for independent samples. For all mean evaluations, a significance level of *p* < 0.05 was used (IBM SPSS Statistics for Windows (version 27); 2021; IBM Corp, New York, NY, USA).

## 3. Results

### 3.1. Performance of UF Process

Figure 1 presents the evolution of the UF fluxes over time. The results are the average of two concentration trials. It is clear that higher filtration flux was obtained during the concentration of goat CW. In both cases, average fluxes are of the order of 30 and 20 Lm^−2^h^−1^, respectively for goat and sheep CW, which indicates the feasibility of the operation of ultrafiltration units in small-scale cheese plants, processing volumes of ca. 500–1000 L milk per day. The figure also presents the calculated evolution of the volumetric concentration factor during the concentration process. This information indicates that it is possible to concentrate whey to obtain high levels of protein and/or fat with the objective to obtain the desired levels of these nutritional components, namely protein, in the LWCs.

### 3.2. Physicochemical Characteristics of LWCs and Kefir Samples

The physicochemical characteristics of the original whey used in the trials and of the LWCs used in the production of kefir samples are displayed in Table 1. It is evident the higher protein content of the sheep LWC, while in the case of fat, the goat LWC presented higher values. It has to be referred that, despite its higher fat content, goat cheese whey had higher UF fluxes than sheep whey. Hence, it can be concluded that the protein level of the feed had a major impact on UF performance.

The physicochemical characteristics of the different kefir formulations produced are displayed in Figure 2. As expected, the products originating with sheep LWC present higher levels of protein, which are nearly double the ones of the products obtained from goat whey. Conversely, goat whey products present higher levels of fat, which results from the higher fat level in the original goat CW.

Figure 3 presents the evolution of the pH and titratable acidity of kefir products over 30 days of refrigerated storage. It is clear the lower pH and the higher acidity of goat kefir products. The sheep kefir products produced with commercial kefir starter without and with the addition of the probiotic culture (EK and EKABT5) presented the highest pH and the lowest acidity values, while the traditional kefir without or with the addition of the probiotic culture (TK and TKABT5) presented pH values below 4.5 and acidity values of the order of 1.3% lactic acid, immediately after production. The acidity increased to values higher than 1.5 on days 10 and 20 and presented a sharp increase from the 20th to the 30th day of storage. In the case of goat kefir samples, only TKABT5 surpassed values of 1.5% lactic acid. The use of the probiotic culture had impacts on these parameters, decreasing the pH and increasing the titratable acidity, both in sheep and in goat kefir products.

Figure 4 shows the elastic (G′) and viscous (G″) moduli as well as the complex viscosity of sheep and goat kefir products. Sheep kefir products presented a solid-like texture while goat kefir products were liquid. Regarding sheep kefir products, the highest values of G′ were obtained for TK and TKABT5 (Figure 4A). EK and EKABT5 presented G′ values ca. one log cycle lower as compared to the ones produced with the traditional kefir. These observations are confirmed by the evaluation of the complex viscosity (Figure 4B). In all the goat kefir products, the viscous modulus (G″) was higher than the elastic modulus (G′) (Figure 4C), indicating the liquid nature of the products, which is also confirmed by their complex viscosity (Figure 4D).

The texture evaluation performed on sheep kefir products is displayed in Figure 5. It is clear the tendency for the increase in the hardness of sheep kefir products between the first and the 30th days of storage. Most probably, this result could be due to the increased links in the protein matrix of the products and/or to the separation of water from the protein matrix (syneresis), which increased the hardness of the protein/lipid matrix.

Figure 6 shows the evolution of the color parameters of all kefir samples over storage. It can be observed that both sheep and goat kefir products show a slight increase in the lightness parameter L* over time (Figure 6A,B). Goat kefir products presented higher values for parameter a* (red-green axis) as compared to sheep kefir products (Figure 6B,C). However, the main difference between sheep and goat kefir products was observed in parameter b* (blue-yellow axis), with goat products presenting values of the order of 4, increasing to 6 on the 30th day of storage, while the lowest value registered for sheep samples was of the order of 6. This observation reflects the more intense yellow color for these samples. Appendix A indicate the values of ΔEab* between different products and for the same product over storage. In most cases these values are higher than 1, indicating that color differences can be detected by a common observer. Fewer differences were observed between kefir products based on goat LWC (ΔEab* values <1).

### 3.3. Microbiological Characteristics of Kefir Products

Regarding the evolution of the microbial composition, Figure 7 compares the counts of lactobacilli, lactococci, and yeasts of the different products during storage. In all cases, the products presented counts of lactobacilli, lactococci, and yeasts of the order of, or higher than, log 7 CFU/g, indicating the good adaptation of LAB and yeasts to the LWC, and the probiotic potential of the products over the storage period.

Concerning the sensory evaluation of the different products, it is clear that further work must be undertaken in order to improve the acceptability of the products. Considering the results presented in Table 2, it can be concluded that sheep kefir products were below an acceptable score of 6, regarding the aroma and taste of samples S-TK and S-TKABT5. Concerning goat products, both G-EK and G-EKABT5 presented unacceptable scores for aroma, taste, and texture. The ranking test allowed the separation of samples into different groups. Regarding sheep products, samples S-EK and S-EKABT5 were ranked 1st and 2nd respectively, being S-TK and S-TKABT5 ranked 3rd and 4th. The opposite pattern was observed with goat kefir samples which were ranked in the following order: 1st G-TKABT5; 2nd G-TK; 3rd G-EKABT5; 4th G-EK.

## 4. Discussion

Considering the performance of the UF process, the direct concentration of sheep or goat CW can lead to the production of LWCs with interesting nutritional properties, which can subsequently be used in the production of fermented synbiotic dairy products with excellent nutritional characteristics. As expected, at higher volumetric concentration factors, the increased solids concentration of retentates started to affect UF filtrate fluxes negatively. This was particularly evident in the case of sheep whey due to its higher protein content. One must also point out the fact that the total solids concentration factors of the retentates do not reflect the VCF 20 aimed. This is due to the dilution of the volume of product with approximately 15 L of water retained in dead volumes at the beginning of the operation and, the need to push the final volume of retentate with the introduction of water in the system. Both these situations originate a certain amount of dilution of the feed and of the retentate, with higher impacts when working in batch mode and when low feed volumes are used. This was particularly evident in the case of goat whey. The characteristics of the UF processing unit used can fit the needs of small-scale dairy industries that, currently, do not possess the capacity to valorize whey. Macedo and coworkers [21] had already reported that an integrated process of ultrafiltration (UF)/nanofiltration (NF) is economically viable in small/medium-sized cheese dairies processing of ca. 3500 L milk/day and they observed that UF membranes of 10 kDa (the same pore size that we used) have higher permeate fluxes than membranes of 1 kDa. The efficiency of the process performed in the present work allows us to consider that the process is feasible even in smaller scale units (i.e., processing ca. 500 L milk/day). Higher protein contents in UF liquid concentrates from sheep cheese whey could be obtained by using diafiltration and adding calcium chloride as has been reported by Pavoni et al. [14]; however, these operations increase the complexity of the valorization process.

It is also important to remark that as a result of the UF process with a VCF of 20 a high volume of whey permeate is obtained (ca. 95 % of the original feed volume). This UF permeate contains most of the lactose of the original whey which is responsible for its high BOD. Although UF concentration allows for the obtention of a valuable product (i.e., LWC), this operation does not solve completely the environmental problem associated with cheese whey. UF whey permeates must be treated by physicochemical treatment (e.g., nanofiltration) or fermentation in order to overcome environmental problems. The application of NF to UF whey permeates allows for the obtention of lactose concentrates and reduces the environmental impact, as has been reported in goat whey treatment [21].

Regarding the physicochemical characteristics of the kefir samples obtained, one can refer that they reflect the composition of the LWCs obtained. The higher values of protein reached in sheep LWC in relation to goat LWC influenced the rheological properties (G′, G″, η*) of the kefir. The lower levels of protein in goat kefir was the cause for their liquid nature. Marnotes et al. [23] also observed that sheep yogurts before freezing were solid, and goat yogurts were viscous liquids due to differences in protein concentrations of the LWCs used as raw materials.

Some differences in kefir from sheep and goats have also been reported. Guangsen et al. [46] compared the kefir produced with ovine and caprine milk and observed significantly higher values of pH in the parameter b* of ovine kefir.

The high counts of lactobacilli, lactococci, and yeasts (>log 7 CFU/g) indicate the good adaptation of microorganisms to the ovine´s and goat´s LWCs. Commercial and traditional kefir did not show particular differences regarding their microbial counts. Several studies compared the microbiological characteristics of kefir produced by commercial cultures or grains [46,47,48,49]. Guangsen et al. [46] report that the counts of lactic acid bacteria were higher in the sheep kefir than in goat kefir. These authors indicated that the sheep kefir was found better than goat kefir due to the microbiological characteristics, volatile compounds, and sensory profile. Biçer et al. [47] evaluated the bacterial microbiota of five commercial and one traditional kefir and observed that the microbial diversity in traditional kefir is higher than in commercial kefir, being *Lactobacillaceae* and *Streptococaccceae*, the most important families in traditional kefir. Guclu et al. [48] studied industrial kefirs (plain kefirs (without fruits) and fruit kefirs) and kefir fermented using kefir grains. They observed counts of total aerobic mesophilic bacteria in the order of log 5.5–8.0 CFU/mL in plain kefir and of log 6.2–8.5 CFU/mL in fruit kefir; higher counts were found in kefir fermented with grains. They did not find any differences in the counts of lactic acid bacteria in plain and fruit kefir measured at two different times of storage. Wang et al. [49] observed that the microbiological and sensory characteristics of the kefir fermented by a compound strain starter and by kefir grains were similar.

Kef & Arslan [34] observed in cow’s milk kefir and goat milk kefir a decrease of streptococci and lactobacilli counts during the storage of the products, slowly until the 7th day and quickly until the 14th day of storage. These results are different than those found in our work since a decrease in the microbiological counts during the 30 days of storage was not observed.

In relation to sensory analysis, it has been reported that the ovine kefir is more appreciated than the caprine kefir [46]. The sensory scores of our ovine and goat kefirs showed significant differences, but some sensorial aspects of both kefirs can be improved. It must be referred that the composition of the LWCs can be tailored according to the needs, particularly concerning the levels of solids, protein, and fat desired for the fermented products. It has also to be referred that, other ingredients such as fruit syrups or jams can be used in the formulations according to consumer’s preferences. Therefore, there is a large margin to improve the sensory acceptability of the products. The improvement of the acceptability of the products will largely increase the feasibility of this approach, envisaging the valorization of CW through the production of fermented dairy products with potentially positive health effects. Furthermore, these products can have a shelf-life of ca. 30 days, which is larger than the one of conventional whey cheeses traditionally produced by small-scale dairy companies.

## 5. Conclusions

Considering the feasibility of the use of UF concentration of CW in small/medium scale cheese plants processing small ruminant’s milk, the present work proves that this operation can allow for the valorization of such byproducts, reducing therefore the environmental impact of the operation and allowing for the development of functional foods with proven health benefits. However, this does not solve the environmental problem associated with cheese whey due to the high amount of permeate obtained, which must be further processed. Despite the need to improve the sensory acceptability of the kefir products developed, it can be considered that this approach represents an excellent opportunity for such companies, which in most cases face the production of CW as a problem and do not take advantage of the potential opportunities to valorize it.

## Figures and Tables

**Figure 1 membranes-13-00473-f001:**
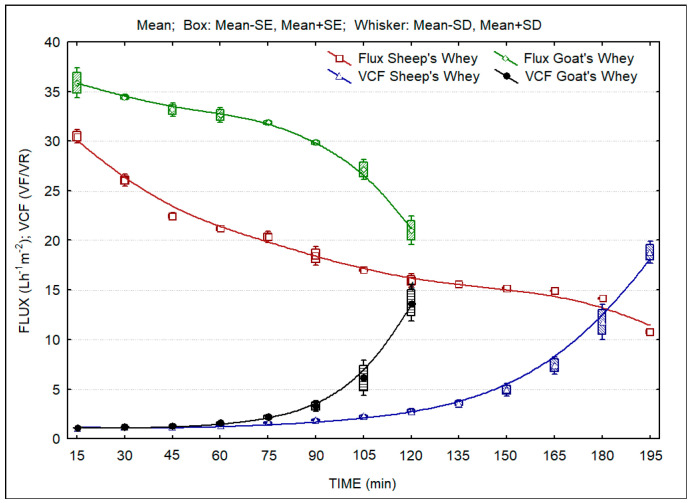
Evolution of ultrafiltration fluxes and volumetric concentration factor during concentration of sheep and goat whey. Fit by distance weighted least squares.

**Figure 2 membranes-13-00473-f002:**
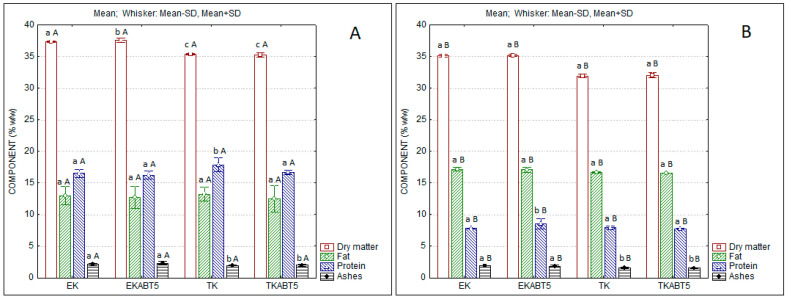
Composition of kefir samples produced with sheep (**A**) and goat (**B**) liquid whey concentrates. (Mean; Standard Deviation-SD). Different lowercase letters a, b, and c indicate significant differences (*p* < 0.05) among formulations produced with the same LWC. Different capital letters A and B indicate differences (*p* < 0.05) between each formulation produced with different LWC (sheep or goat). EK = Commercial kefir; EKABT5 = Commercial kefir + probiotic culture; TK = Traditional kefir; TKABT5 = Traditional kefir + probiotic culture.

**Figure 3 membranes-13-00473-f003:**
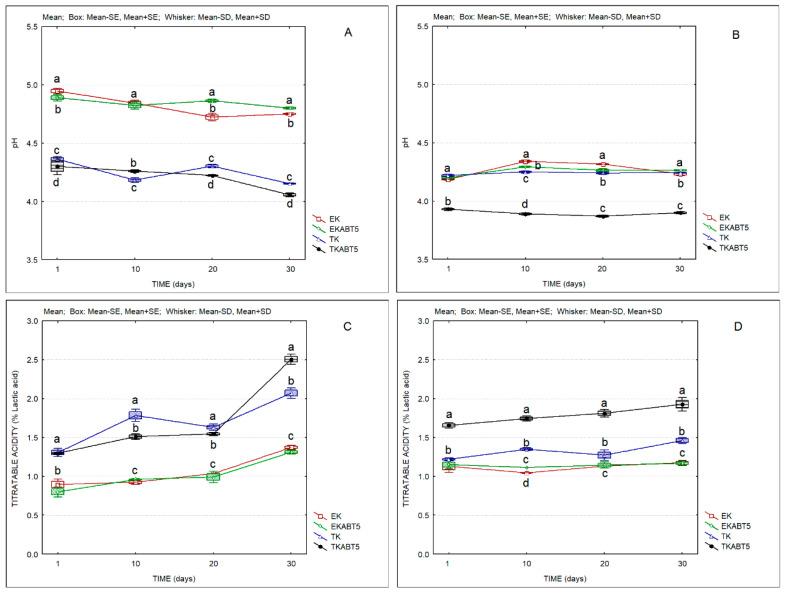
pH and titratable acidity of kefir samples produced with sheep and goat liquid whey concentrates. (**A**) pH of sheep kefir samples; (**B**) pH of goat kefir samples; (**C**) Titratable acidity of sheep kefir samples; (**D**) Titratable acidity of goat kefir samples. (Average values; Standard Error-SD; Standard Deviation-SD). EK = Commercial kefir; EKABT5 = Commercial kefir + probiotic culture; TK = Traditional kefir; TKABT5 = Traditional kefir + probiotic culture. Different lowercase letters a, b, and c indicate significant differences (*p* < 0.05) among formulations produced with the same LWC.

**Figure 4 membranes-13-00473-f004:**
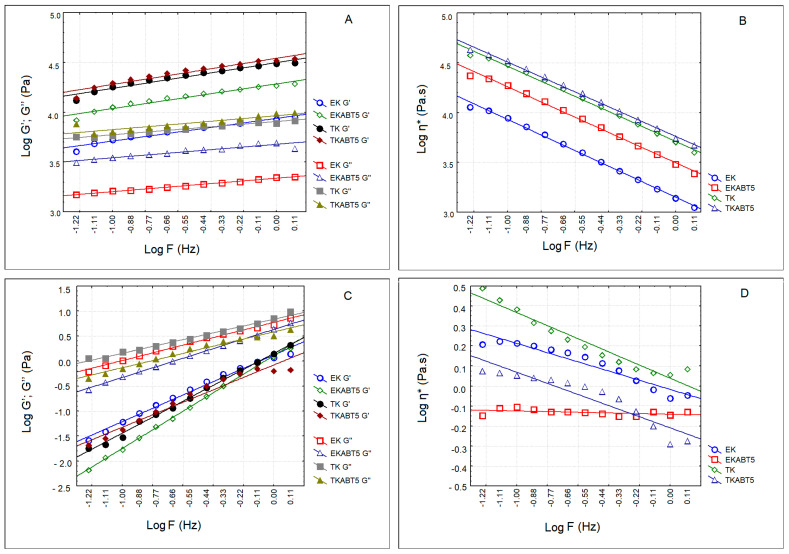
Elastic (G′) and viscous moduli (G″) and complex viscosity (η*) of kefir samples produced with sheep and goat liquid whey concentrate. (**A**) G′ and G″ of sheep kefir samples; (**B**) η* of sheep kefir samples; (**C**) G′ and G″ of goat kefir samples; (**D**) η* of goat kefir samples. EK = Commercial kefir; EKABT5=Commercial kefir + probiotic culture; TK = Traditional kefir; TKABT5 = Traditional kefir + probiotic culture.

**Figure 5 membranes-13-00473-f005:**
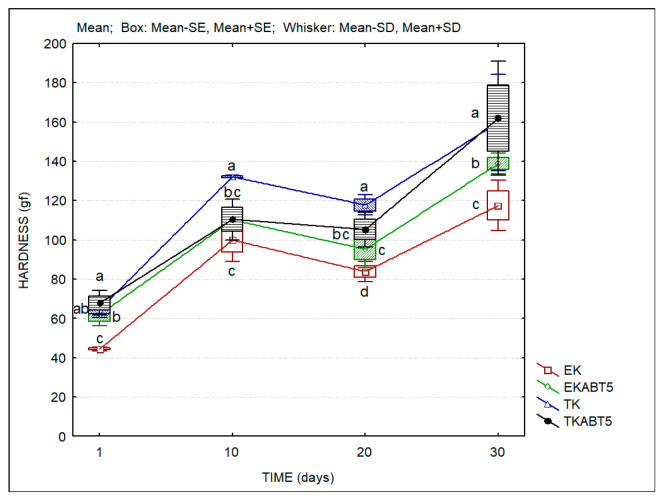
The hardness of kefir samples produced with sheep liquid whey concentrates. (Average values; Standard Error-SD; Standard Deviation-SD). Different lowercase letters a, b, c, and d indicate significant differences (*p* < 0.05) among the days of storage. EK = Commercial kefir; EKABT5 = Commercial kefir + probiotic culture; TK = Traditional kefir; TKABT5 = Traditional kefir + probiotic culture.

**Figure 6 membranes-13-00473-f006:**
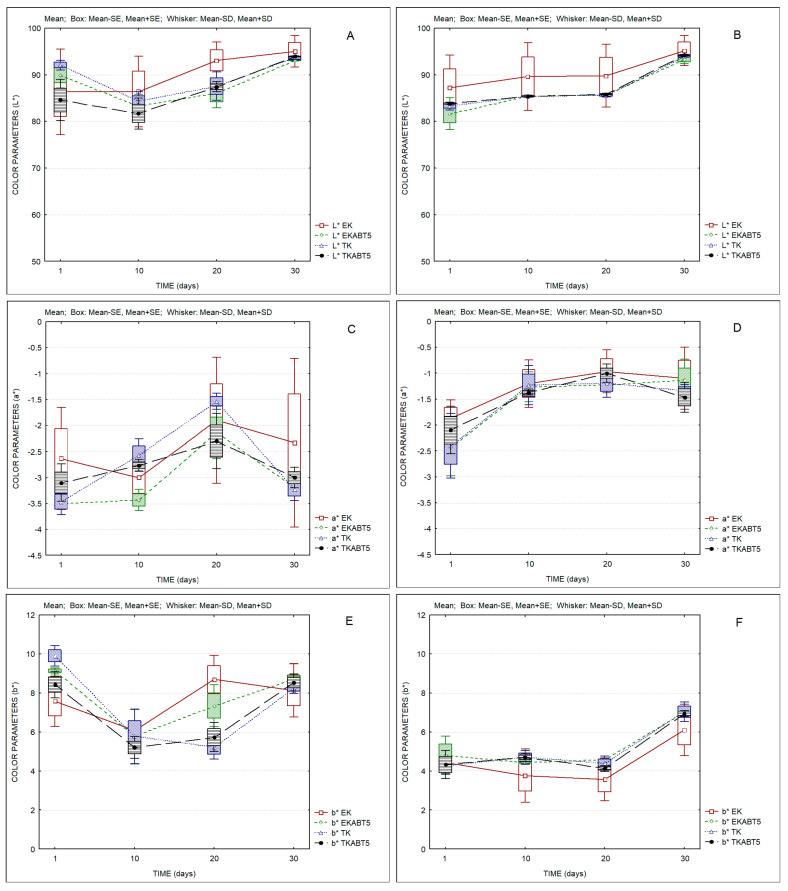
Color parameters of kefir samples produced with sheep and goat liquid whey concentrate. (**A**) L* of sheep kefir samples; (**B**) L* of goat kefir samples; (**C**) a* of sheep kefir samples; (**D**) a* of goat kefir samples; (**E**) b* of sheep kefir samples; (**F**) b* of goat kefir samples. (Average values; Standard Error-SD; Standard Deviation-SD). EK = Commercial kefir; EKABT5 = Commercial kefir + probiotic culture; TK = Traditional kefir; TKABT5 = Traditional kefir + probiotic culture.

**Figure 7 membranes-13-00473-f007:**
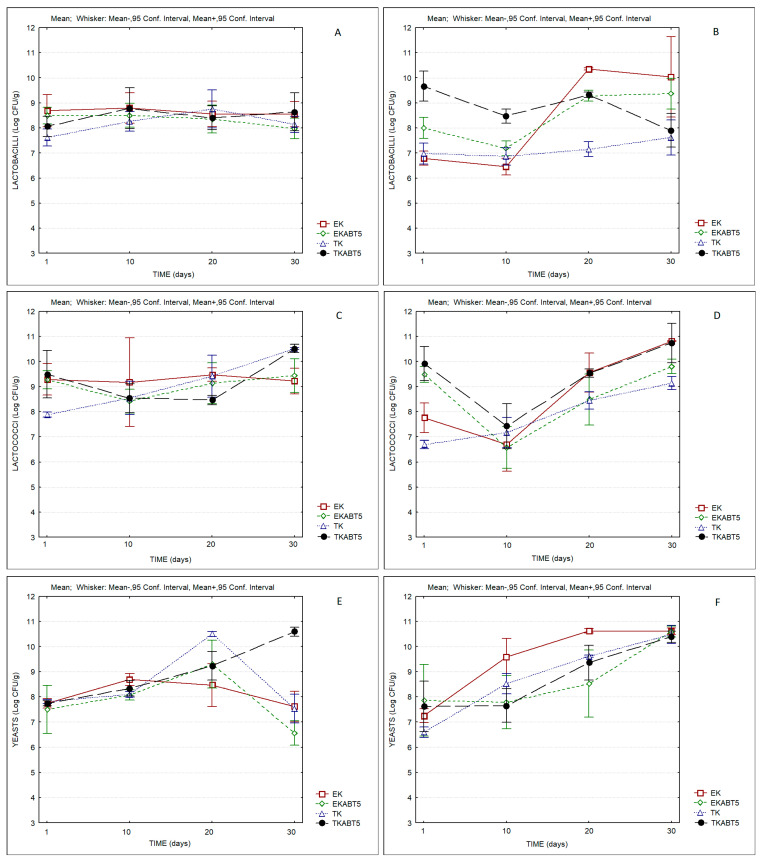
Microbial characteristics of kefir samples produced with sheep and goat liquid whey concentrate. (**A**) Lactobacilli counts of sheep kefir samples; (**B**) Lactobacilli counts of goat kefir samples; (**C**) Lactococci counts of sheep kefir samples; (**D**) Lactococci counts of goat kefir samples; (**E**) Yeasts counts of sheep kefir samples; (**F**) Yeasts counts of goat kefir samples (Average values and 0.95 confidence interval). EK = Commercial kefir; EKABT5 = Commercial kefir + probiotic culture; TK = Traditional kefir; TKABT5 = Traditional kefir + probiotic culture.

**Table 1 membranes-13-00473-t001:** Composition of sheep and goat whey and of liquid whey concentrates (LWCs).

Products	Dry Matter (%)	Protein (%)	Fat (%)	Ash (%)
Sheep whey	8.63 ± 0.68	1.32 ± 0.09	1.18 ± 0.43	2.15 ± 0.27
Sheep LWC	34.74 ± 0.68	16.35 ± 0.36	14.12 ± 0.50	1.98 ± 0.67
Goat whey	6.61 ± 0.09	0.73 ± 0.01	1.76 ± 0.47	1.03 ± 0.24
Goat LWC	32.23 ± 0.10	7.84 ± 0.45	19.32 ± 0.12	1.55 ± 0.07

**Table 2 membranes-13-00473-t002:** Sensory scores of kefir samples produced with sheep and goat LWCs.

Products	S-Aroma	S-Color	S-Taste	S-Texture
S-EK	5.9 ± 1.5 ^aA^	7.0 ± 1.6 ^aA^	6.3 ± 2.0 ^aA^	7.1 ± 1.7 ^aA^
S-EKABT5	6.0 ± 1.8 ^aA^	7.0 ± 1.5 ^aA^	6.3 ± 2.0 ^aA^	7.2 ± 1.4 ^aA^
S-TK	4.5 ± 1.5 ^bA^	6.8 ± 1.6 ^aA^	4.1 ± 1.9 ^bA^	5.6 ± 2.0 ^bA^
S-TKABT5	4.8 ± 1.9 ^bA^	6.8 ± 1.7 ^aA^	3.8 ± 1.8 ^bA^	5.6 ± 1.9 ^bA^
G-EK	4.9 ± 2.0 ^abB^	7.5 ± 1.6 ^aA^	3.7 ± 2.3 ^aB^	5.2 ± 1.7 ^aB^
G-EKABT5	4.7 ± 2.0 ^aB^	7.6 ± 1.7 ^aA^	3.1 ± 1.8 ^aB^	5.6 ± 2.1 ^abB^
G-TK	5.8 ± 1.7 ^abB^	7.9 ± 1.3 ^aB^	5.5 ± 2.2 ^bB^	6.5 ± 1.6 ^bB^
G-TKABT5	6.0 ± 2.0 ^bB^	8.0 ± 1.1 ^aB^	5.8 ± 2.1 ^bB^	6.4 ± 2.0 ^abA^

Different superscript letters a and b indicate significant differences between formulations produced with the same LWC. Different superscript capital letters A and B indicate differences between each formulation produced with different LWCs.

## Data Availability

Data is included in the manuscript or in Appendix A.

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
