# Peer review of "Application of Ultrafiltration to Produce Sheep’s and Goat’s Whey-Based Synbiotic Kefir Products"

_membranes, 2023, doi:10.3390/membranes13050473_

Round 1
Reviewer 1 Report
This study focuses at the use of UF to concentrate liquid whey from ovine and caprine source, in order to develop kefir products. Overall, this is an interesting study where the main focus in on the use of different starters. UF-concentration is performed using a 10 kDa membrane, up to a 20X volumic concentration factor. Therefore, 25L of concentrate is prepared, leaving behinf 475L of whey permeate. This is an important environmental issue since it contains most of the lactose which is responsible for the high Biochemical Oxygen Demand (BOD) of whey (ca. 40-50 g/L).
The authors absolutely need to notify this drawback (or limitation) in their discussion/ conclusion: UF-concentration of whey improves the use of whey proteins but it does not solve the environmental problem associated with cheese whey.
Reviewer 2 Report
Title:
Application of ultrafiltration to produce sheep's and goat's whey based synbiotic kefir products.
General comments:
The paper still needs English revisions
Specific comments:
Abstract:
Why the abstract is divided into several sections? Is it the journal guidelines? Could you please make the abstract as one section with no subtitle for each part?
Background: Membrane filtration technologies are the best available tools to manage dairy 13
byproducts such as cheese whey (CW) and buttermilk. These technologies allow for the selective 14
concentration of specific components of such products, namely proteins. Due to their acceptable 15
costs and ease of operation, these technologies can be applied by small/medium scale dairy plants. 16
Therefore, these companies can transform byproducts such as CW into value-added dairy products. 17
Methods: In the present work, sheep’s and goat’s CW were concentrated by ultrafiltration (UF) 18
envisaging their utilization in the production of synbiotic dairy products based on kefir and 19
probiotic bacteria, with inulin. For each type of whey, four formulations based on a commercial kefir 20
starter or a traditional kefir, without or with the addition of a probiotic culture, were produced. All 21
samples were evaluated regarding their physicochemical, microbiological and sensory properties. 22
Results: The UF operation performed with a pilot plant allowed to conclude that this process can be 23
applied even in dairy plants processing ca.500 L/milk/day envisaging the production of liquid whey 24
concentrates (LWCs) that can be used as the main ingredient of the formulations. The 25
physicochemical characteristics of the kefir samples produced reflect the composition of the LWCs. 26
Sheep’s and goat’s products presented high levels of protein (>15% for sheep’s and >7.5% for goat´s 27
products). The fat content of sheep’s products was significantly lower than that of goat’s products. 28
All samples presented counts of lactic acid bacteria higher than log 7 CFU/mL, indicating the good 29
adaptation of microorganisms to the matrixes. The sensory evaluation indicated that further work 30
must be undertaken in order to improve the acceptability of the products. 31
Conclusions: It could be concluded that small/medium scale dairy plants can use UF equipment 32
with the aim to valorize sheep´s and goat´s CW through the production of dairy formulations with 33
added prebiotics and probiotics, while reducing the environmental impact of their activities.
Could you please show more date in the abstract?
Introduction:
Please extend the second paragraph in the introduction to have more info about membrane filtration, UF, and the engineering behind that and studies who worked on that before!
Material and methods:
No number should start the sentence, e.g. 500 L
Results and discussions:
Why didn’t you standardize the composition of both whey before the UF?
Table 1 shows that the protein content was not standardized in the whey of sheep and goats that’s why you have different composition in the final concentrates.
Based on that you will find different characteristics!
There are many work done on UF that were not mentioned in the paper!!
The paper needs English revisions.
Round 2
Reviewer 2 Report
Thanks to the authors for applying my comments. All the best!